# Prioritising cardiovascular disease risk assessment to high risk individuals based on primary care records

Ryan Chung[1]*, Zhe Xu[1], Matthew Arnold[1¤], David Stevens[1], Ruth Keogh[2], Jessica Barrett[3], Hannah Harrison[4], Lisa Pennells[1], Lois G. Kim[1,5], Emanuele DiAngelantonio[1,5,6,7,8], Ellie Paige[9], Juliet A. Usher-Smith[10], Angela M. Wood[1,5,6,7,11]

1 British Heart Foundation Cardiovascular Epidemiology Unit, Department of Public Health and Primary Care, University of Cambridge, Cambridge, United Kingdom, 2 London School of Hygiene and Tropical Medicine, Faculty of Epidemiology & Population Health, London, United Kingdom, 3 Medical Research Council Biostatistics Unit, University of Cambridge, Cambridge, United Kingdom, 4 Centre for Cancer Genetic Epidemiology, Department of Public Health and Primary Care, University of Cambridge, Cambridge, United Kingdom, 5 National Institute for Health and Care Research Blood and Transplant Research Unit in Donor Health and Behaviour, University of Cambridge, Cambridge, United Kingdom, 6 British Heart Foundation Centre of Research Excellence, University of Cambridge, Cambridge, United Kingdom, 7 Health Data Research UK Cambridge, Wellcome Genome Campus and University of Cambridge, Cambridge, United Kingdom, 8 Health Data Science Research Centre, Human Technopole, Milan, Italy, 9 National Centre for Epidemiology and Population Health, Australian National University, Canberra, Australia, 10 Primary Care Unit, Department of Public Health and Primary Care, University of Cambridge, Cambridge, United Kingdom, 11 Cambridge Centre of Artificial Intelligence in Medicine, Cambridge, United Kingdom

¤ Current address: AstraZeneca PLC, Cambridge, United Kingdom
* rkyc2@medschl.cam.ac.uk

## Abstract

### Objective

To provide quantitative evidence for systematically prioritising individuals for full formal cardiovascular disease (CVD) risk assessment using primary care records with a novel tool (eHEART) with age- and sex- specific risk thresholds.

### Methods and analysis

eHEART was derived using landmark Cox models for incident CVD with repeated measures of conventional CVD risk predictors in 1,642,498 individuals from the Clinical Practice Research Datalink. Using 119,137 individuals from UK Biobank, we modelled the implications of initiating guideline-recommended statin therapy using eHEART with age- and sex-specific prioritisation thresholds corresponding to 5% false negative rates to prioritise adults aged 40–69 years in a population in England for invitation to a formal CVD risk assessment.

### Results

Formal CVD risk assessment on all adults would identify 76% and 49% of future CVD events amongst men and women respectively, and 93 (95% CI: 90, 95) men and 279 (95% CI: 259, 297) women would need to be screened (NNS) to prevent one CVD event. In contrast, if

**Data Availability Statement:** This research has been conducted using the UK Biobank Resource under Application Number 26865. Data from the

Clinical Practice Research Datalink (CPRD) were obtained under licence from the UK Medicines and Healthcare products Regulatory Agency (protocol 162RMn2). Supplementary code for eHEART is available online (www.github.com/ryanchung894/eHEART).

**Funding:** This work was supported by core funding from the: British Heart Foundation (RG/13/13/30194; RG/18/13/33946), BHF Cambridge Centre of Research Excellence (RE/13/6/30180) and NIHR Cambridge Biomedical Research Centre (BRC-1215-20014) [*]. *The views expressed are those of the author(s) and not necessarily those of the NIHR, NHSBT or the Department of Health and Social Care. This work was funded by the Medical Research council (MR/K014811/1). The study funders played no role in the design, analysis or interpretation of the study. R.C. is funded by a BHF PhD studentship (FS/18/56/34177). Z.X. is funded by the Chinese Scholarship Council. M.A. was funded by a British Heart Foundation Programme Grant (RG/18/13/33946). D.S. was funded by the NIHR Cambridge Biomedical Research Centre (BRC-1215-20014) [*]. J.B. was funded by a Medical Research Council fellowship (MR/L501566/1) and unit programme (MC_UU_00002/5). L.G.K. was funded by the NIHR BTRU in Donor Health and Genomics (NIHR BTRU-2014–10024) and is funded by the NIHR BTRU in Donor Health and Behaviour (NIHR203337) [*]. J.A.U. is funded by an NIHR Advanced Fellowship (NIHR300861). A.M.W. is part of the BigData@Heart Consortium, funded by the Innovative Medicines Initiative-2 Joint Undertaking under grant agreement No 116074. A.M.W. is supported by the BHF-Turing Cardiovascular Data Science Award (BCDSA \100005). Cambridge Service for Data Driven Discovery (CSD3) was provided by Dell EMC and Intel using Tier-2 funding from the Engineering and Physical Sciences Research Council (capital grant EP/P020259/1), and DiRAC funding from the Science and Technology Facilities Council (www.dirac.ac.uk). There was no additional external funding received for this study The funder provided support in the form of salaries for authors M.A., but did not have any additional role in the study design, data collection and analysis, decision to publish, or preparation of the manuscript.

**Competing interests:** During the drafting of the manuscript, M.A. became an employee of AstraZeneca. This does not alter our adherence to PLOS ONE policies on sharing data and materials.

eHEART was first used to prioritise individuals for formal CVD risk assessment, we would identify 73% and 47% of future events amongst men and women respectively, and a NNS of 75 (95% CI: 72, 77) men and 162 (95% CI: 150, 172) women. Replacing the age- and sex-specific prioritisation thresholds with a 10% threshold identify around 10% less events.

## Conclusions

The use of prioritisation tools with age- and sex-specific thresholds could lead to more efficient CVD assessment programmes with only small reductions in effectiveness at preventing new CVD events.

## Introduction

The WHO estimate that the majority of premature cardiovascular disease (CVD) events are preventable through lifestyle choices and pharmacotherapy that target modifiable risk factors [1]. Many countries implement population-wide CVD risk assessment programmes in primary care that recommend use of established risk prediction tools to identify individuals at high risk of CVD and guide patient and clinical decision-making [2–8]. More recently, to reduce programme running costs and address rising health inequalities in access to preventative care, guidelines and policy makers have advocated using primary care records for systematic prioritisation of individuals *before* formal CVD risk assessment [7]. This has become even more important given the current COVID-19 related burdens on national health care provision. As an example, in addition to recommendations to formally assess all individuals in England, individuals are recommended to be prioritised for a formal CVD risk assessment if their estimated risk, using existing data from primary care health records, is greater than 10% risk over ten years [7]. However, quantitative evidence of the health impact of prioritisation when using a fixed 10% threshold in clinical practice is limited, and no specific risk tool is recommended for systematic prioritisation of risk.

Therefore, to help in the formulation of evidence-based CVD risk assessment programmes, we aim to evaluate two key aspects. First, we compare the *population-wide* formal CVD risk assessment approach against approaches that *systematically prioritise* individuals before full formal CVD risk assessment. Here, we evaluate two distinct prioritisation tools based on primary care records: a novel tool (eHEART), developed in this manuscript to leverage the sparse and sporadically observed longitudinal data on conventional CVD risk factors recorded primary care records; and the other, being an existing CVD risk tool (QRISK2) [9], that has been derived on 2.3 million individuals from the QResearch database and which implements a substitution approach for any missing conventional risk factors based on age, sex and ethnicity. Second, we compare the use of a fixed 10% prioritisation threshold versus age- and sex-specific prioritisation thresholds.

## Methods

### CPRD data source

To derive eHEART we used anonymised person-level primary care records from general practices contributing data to Clinical Practice Research Datalink GOLD (CPRD) with linked information on hospital episodes from Hospital Episode Statistics (HES), and death registrations from the Office of National Statistics (ONS) [10]. Primary care records were extracted

for individuals from the latest of: date of their registration at general practice plus 6 months, their 30[th] birthday, date when the general practice provided "up-to-standard" data [11], or 1[st] April 2004 (the date of introduction of the Quality and Outcomes Framework) [12], until the earliest of: date of their first (i.e. "incident") newly recorded CVD event, date of de-registration at the general practice, their 85[th] birthday, date of death, last contact date for the practice with CPRD, or 31 May 2019 (the end of data availability). All analyses focus on individuals without known pre-existing CVD and without prescribed statins. The data used in this study were obtained under licence from the UK Medicines and Healthcare Products Regulatory Agency (protocol 162RMn2).

## UK Biobank data source

Data from UK Biobank, a prospective cohort study with detailed baseline information and linked primary care record data (from the The Phoenix Partnership (TPP), Egton Medical Information Systems (EMIS) and Vision GP system suppliers) for 177,361 individuals [13, 14], was used to model the implications of prioritising individuals for formal CVD assessment and subsequent initiation of guideline-recommended statin therapy in a primary care setting. UK Biobank was chosen due to the complete data available at baseline to replicate being able to perform a full risk assessment. Population health modelling focussed on individuals without known pre-existing CVD and without prescribed statins at baseline. This research has been conducted using the UK Biobank Resource under Application Number 26865.

## Definition of CVD outcome

We ascertained individuals with a first ever incident CVD, defined as nonfatal or fatal events of coronary heart disease (CHD) (including myocardial infarction and angina), stroke, and transient ischemic attack, as those with a relevant Read-code or ICD-10 code (listed in Table 1 in S2 File) appearing in the primary care data, hospital episodes (main or secondary diagnostic code position in the admitted patient care component of the Hospital Episode Statistics data), or death registry (underlying or contributing cause of death) during follow-up.

## CVD risk predictors

Conventional and commonly recorded CVD risk predictors [5, 15] in primary care were pre-selected for inclusion into the estimated eHEART prioritisation tool and included: age (in years), sex (male or female), diabetes status (yes or no), hypertension medication (yes or no), systolic blood pressure (SBP) (mmHg), total cholesterol (mmol/litre) and high-density lipo-protein (HDL) cholesterol (mmol/litre) and smoking status (current smoker or not) (details of measurements have been previously described) [16].

The following measurements were considered biologically implausible and were changed to missing ($\sim$ 0.4% of measurements): SBP <60 or >250 mmHg; total cholesterol <1.75 or >20 mmol/litre; and HDL cholesterol <0.3 or >3.1 mmol/litre [17].

## Statistical analysis

**The eHEART prioritisation tool.**   To optimise the nature of longitudinal primary care records, we used a landmark-age approach to derive 90 age- (ie, 40, 41, 42, . . .to 84 years) and sex-specific Cox models as previously described [18]. Each model utilised primary care records from persons alive without pre-existing CVD and without prescribed statins at the landmark-age by using available risk predictors recorded before the landmark-age to predict 10-year risk of incident CVD outcomes (Fig 1 in S1 File).

Each model was developed in two stages. In the first stage, a multivariate mixed-effects model [19] was fitted to repeat measures of smoking status, SBP, total cholesterol and HDL cholesterol recorded before the landmark-age and used to estimate risk predictor values at the landmark-age amongst individuals who had at least one past record of any one predictor. The model included fixed- and correlated random-intercepts and linear age fixed-effects (quadratic terms were not significant) for each predictor, as well as an interaction between SBP and an age-varying binary covariate for hypertension treatment, and an interaction between total cholesterol and an age-varying binary covariate for statin treatment (but excluded those with pre-statins for Cox model). The model intrinsically accounts for missing data in the predictors with the assumption that predictor values from individuals with incomplete data are from the same multivariate normal distribution for predictor values as individuals with observed data. Further details are given in Text 1 in S2 File.

In the second stage, we derived Cox models with time since landmark age as the underlying time scale and the following predictors: diabetes status, treatment for hypertension, and the estimated values of SBP, total cholesterol, HDL cholesterol, and smoking status from the multivariate mixed-effects model. The four estimated predictors were entered as linear terms. The outcome was incident CVD and censoring occurred for individuals at end of their follow-up, either date of CVD event (cases), the study end date or date of death (from non-CVD causes). No violation of the proportional hazards assumption was identified. The Cox model was used to estimate 10-year CVD risks, constituting the "eHEART risk".

**Model internal validation and assessment.**   The eHEART prioritisation model was derived using primary care records from 2/3 of general practices and internally validated on the remaining 1/3 of general practices. In the validation dataset, we calculated measures of predictive accuracy using the Brier score [20] and risk discrimination using the C-index [21]. Calibration was assessed visually by plotting mean predicted risk against mean observed risk by deciles of predicted risk [22, 23] and by assessing the calibration slope, estimated from the linear regression of log transformed observed risk and predicted risk in predicted risk decile groups.

**The QRISK2 formal risk assessment.**   We calculated formal 10-year CVD risks using QRISK2 (recommended by UK NICE guidelines until May 2023) using the values at the baseline assessment of UK Biobank's and published coefficients. Chronic kidney disease, family history and historical prescriptions of statins from primary care records were supplemented in addition to the baseline records, and missing values were imputed as described above.

**Population health modelling.**   We compared the potential public health impact of using eHEART to prioritise individuals for invitation to a formal CVD risk assessment with QRISK2 (Fig 1) versus the whole-population strategy to invite all adults for formal CVD risk assessment. Amongst 174,715 individuals in UK Biobank with linked primary care records prior to baseline, we calculated eHEART and QRISK2 (using complete data available at baseline survey) risk at the individual's age at baseline.

Due to UK Biobank being a cohort of healthier individuals than the UK primary care population, we recalibrated the eHEART and QRISK2 models using age-group- and sex-specific risk factor levels and CVD incidence rates from UK Biobank, and then *rescaled* the predicted eHEART and QRISK2 risks to achieve 10-year CVD risk distributions that would be expected in a UK primary care setting, using methods previously described [23]. Details are provided in Text 2 in S2 File.

We modelled a population of 50,000 men and 50,000 women aged 40–69 years with age profiles matching that of the contemporary UK population (2017 mid-year population) [24], and CVD incidence rates as observed in CPRD individuals without pre-existing CVD. We assumed a policy of statin initiation for individuals at ≥10% predicted QRISK2 10-year risk as recommended by National Institute for Health and Care Excellence (NICE) guidelines [7] and

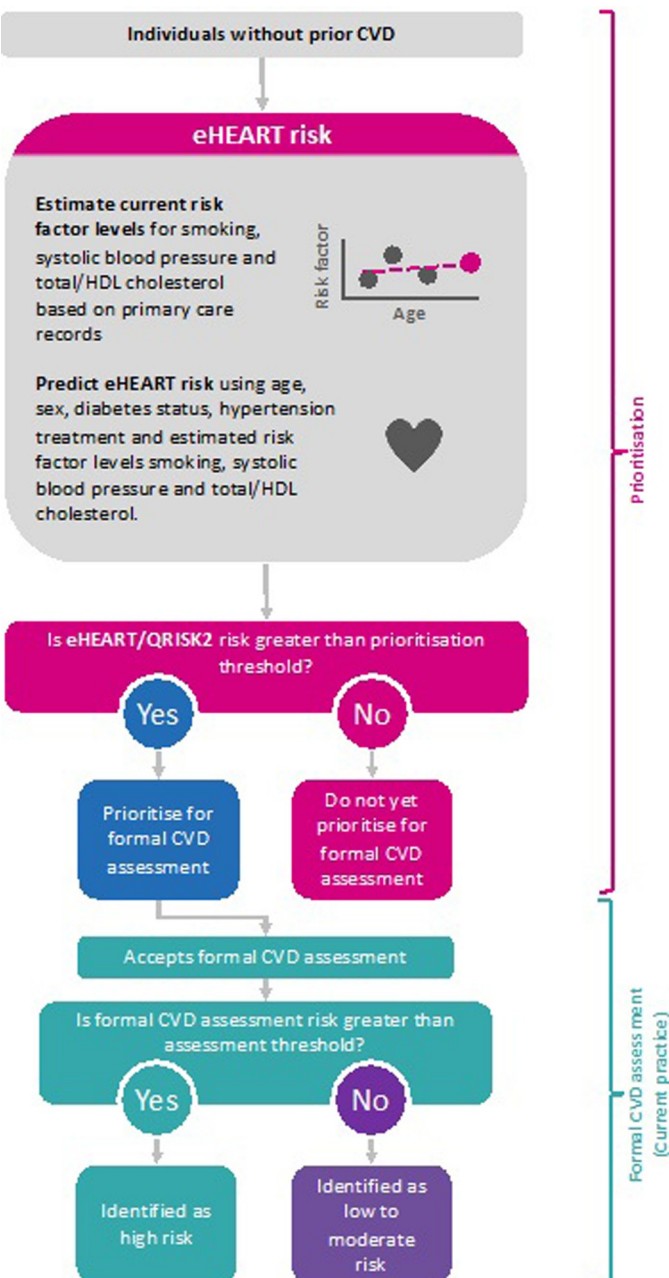

**Fig 1. Flow chart of implementation of eHEART and QRISK2 as a prioritisation tool for formal cardiovascular disease assessments.** Abbreviations: BMI, body mass index; CVD, cardiovascular disease; HDL, high density lipoprotein.

assumed statin allocation would reduce CVD risk by 20% [25]. We assumed 50% statin compliance and a 50% invitation uptake when formally inviting all individuals for a formal assessment. We then modelled the impact of prioritising individuals for QRISK2 formal assessment, based on a fixed prioritisation threshold of ≥10% estimated eHEART 10-year risk as currently recommended by NICE guidelines [7] and also on age- and sex-specific prioritisation thresholds selected to correspond to 5% false negative rates in UK Biobank; these were chosen by ranking the estimated eHEART 10-year risks in UK Biobank individuals who go on to have a CVD

event in the next 10 years. The prioritisation thresholds were chosen as the minimum 10-year risk such that 5% of events would be missed. Population health impact was assessed using the number needed to screen (i.e., undergo formal assessment) to prevent one CVD event, the number of events identified and the number need to invite for formal assessment to prevent one CVD event assuming an increased invitation uptake of 55% after prioritisation [26].

In sensitivity analyses, first we repeated population health-analyses by using QRISK2 (using primary care records) (Text 3 in S2 File) to prioritise individuals for invitation to a formal CVD risk assessment with QRISK2 (using complete data available at baseline-survey). Second, we repeated population-health analyses including all individuals, including those without a primary care record for any one of SBP, HDL, total cholesterol and smoking status. For the eHEART tool, we assumed all these individuals would be directly invited for a formal risk assessment, and for the estimated QRISK2 prioritisation tool, additional missing values were imputed as described above. Third, to assess the use of prioritisation tools at different formal risk assessments thresholds, we repeated the analyses assuming a 5% formal risk assessment threshold in combination with a fixed 5% prioritisation threshold and age- and sex-specific prioritisation thresholds selected to correspond to 2.5% false negative rates in UK Biobank.

Analyses were performed with R x64 3.6.1 [27] and Stata version 15. This study follows the RECORD (Table 2 in S2 File) and TRIPOD (Table 3 in S2 File) reporting guidelines [28, 29]. R code to predict eHEART for new individuals is provided online.

**Patient and public involvement.** Patients and the public were not directly involved in designing or implementing this research project. Written informed consent was obtained from all participants

## Results

### Study population and baseline characteristics using CPRD

We identified 2,154,089 individuals from 398 practices contributing to the CPRD database with linked hospital episode statistics and ONS death registrations in England and who met our inclusion criteria. For our primary analysis we excluded 511,591 (24%) without at least 1 record of SBP, total cholesterol, HDL cholesterol or smoking status, leaving 1,642,498 individuals. A total of 263 practices were randomly assigned to the derivation dataset and the remainder (n = 135) to a validation cohort. Overall, 1,120,053 and 522,445 individuals were included in the derivation and validation cohorts respectively (Fig 2 in S1 File). Compared to those included in our analysis, individuals without any measurement on SBP, cholesterol, or smoking were slightly older, more likely to be male, and had higher CVD incidence rate (Table 1 in S1 File).

The baseline characteristics of individuals are summarised in Table 1 (split by the derivation and validation cohorts in Table 2 in S1 File). Overall, 45% were men and 55% were women; 3% of men and 3% of women had a reported diabetes diagnosis and 16% of men and 23% of women were prescribed anti-hypertension medication before the study entry (Table 1). The number of individuals with CVD risk factors recorded in primary care varied by age and sex (for example, from 21,674 men at age 85 to 188,959 women at age 46) (Figs 3 and 4 in S1 File). There were more repeated SBP than cholesterol measurements recorded (Table 1 and Figs 3 and 4 in S1 File). Characteristics were similar for individuals in the derivation and validation cohorts (Table 2 in S1 File).

In the derivation cohort, 104,830 (6%) individuals had an incident CVD event recorded in either primary care, HES or death registry (Fig 5 in S1 File) over a median 9 years of follow-up (IQR: 5–11). The crude CVD incidence rates per 1000 person years increased with age (from 2.13 in 40-year-olds to 54.88 in those aged 85) and were higher among men than women (Fig

**Table 1. Key characteristics of individuals in CPRD cohort.**

| Characteristics | Men, N = 746,386 | | | Women, N = 896,112 | | |
|---|---|---|---|---|---|---|
| | Mean (SD) or n (%) | No. (%) of persons with value | Median (IQR) of measures per person | Mean (SD) or n (%) | No. (%) of persons with value | Median (IQR) of measures per person |
| Earliest age during follow-up, years | 50.4 (12.7) | 746,836 (100) | - | 51.3 (13.6) | 896,112 (100) | - |
| History of diabetes§ | 21,936 (3.0) | 746,836 (100) | - | 20,662 (3.0) | 896,112 (100) | - |
| Blood pressure-lowering medication prescriptions^ | 119,230 (16.0) | 746,836 (100) | - | 203,394 (22.7) | 896,112 (100) | - |
| Current smoker, n (%)# | 192,509 (25.8) | 438,638 (59) | 3 (2–6) | 176,719 (19.7) | 424,268 (47) | 4 (2–7) |
| Systolic blood pressure mm Hg, mean (SD)# | 136.7 (18.5) | 699,229 (93) | 4 (2–11) | 131.6 (20.3) | 870,075 (97) | 6 (3–14) |
| Total cholesterol mmol/litre, mean (SD)# | 5.4 (1.0) | 532,620 (71) | 2 (1–5) | 5.6 (1.0) | 624,189 (70) | 2 (1–5) |
| HDL cholesterol mmol/litre, mean (SD)# | 1.3 (0.4) | 490,003 (66) | 2 (1–4) | 1.6 (0.4) | 573,842 (64) | 2 (1–4) |

* Included 1,642,498 individuals from Clinical Practice Research Datalink, Hospital Episode Statistics, and the Office for National Statistics, England, United Kingdom, 2004–2019, aged 40–85 years, without prevalent CVD and statin treatment before study entry, and had least 1 measurement on any of systolic blood pressure, total cholesterol, HDL cholesterol, or smoking status between their study entry and study exit dates.

§Defined as ever having recorded diagnosis of diabetes before the study entry

^Defined as ever having recorded use of blood-pressure lowering medications before the study entry

#Proportion or mean (standard deviation) of the first ever measurement

6 in S1 File). In the validation cohort, 32,862 (6%) individuals had an incident CVD event over a median 9 years of follow-up (IQR: 5–11).

## Derivation of the eHEART model

We estimated person-level risk predictor values for smoking status, SBP, HDL and total cholesterol at each age of follow-up; all estimated model coefficients are provided (Table 3 in S1 File). Mean levels of estimated SBP and HDL cholesterol were higher at older ages, as were the proportions of individuals on hypertensive treatment and with diagnosed diabetes. Estimated levels of total cholesterol remained stable across age, whilst the proportion of smokers declined with age (Fig 7 in S1 File).

The eHEART age- and sex-specific hazard ratios for SBP, HDL, total cholesterol, smoking and history of diabetes steadily attenuated towards 1 at older ages, whilst the hazard ratios for hypertensive treatment declined to a lesser extent, especially amongst women (Fig 2). Predicted eHEART risks increased at older ages and changed in distribution from a positive skew to bimodal (due to the contribution of the risk factor for hypertensive treatment) (Fig 8 in S1 File).

## Internal validation of eHEART

In the validation cohort, the overall C-index for eHEART was 0.771 (95% CI: 0.769, 0.772) and was higher in women (C-index = 0.786, 95% CI:0.784, 0.788) than in men (C-index = 0.741, 95% CI: 0.739, 0.742) and higher in younger individuals (Fig 3). The Brier score was also lower (better) in women (0.2354, 95% CI: 0.2340, 0.2368) than in men (0.3085, 95% CI: 0.3068, 0.3102) (Table 4 in S1 File). Calibration plots by decile of eHEART risk showed good agreement with observed risk (Figs 9 and 10 in S1 File). The calibration slope was more deviatory from 1 for older ages (Fig 11 in S1 File).

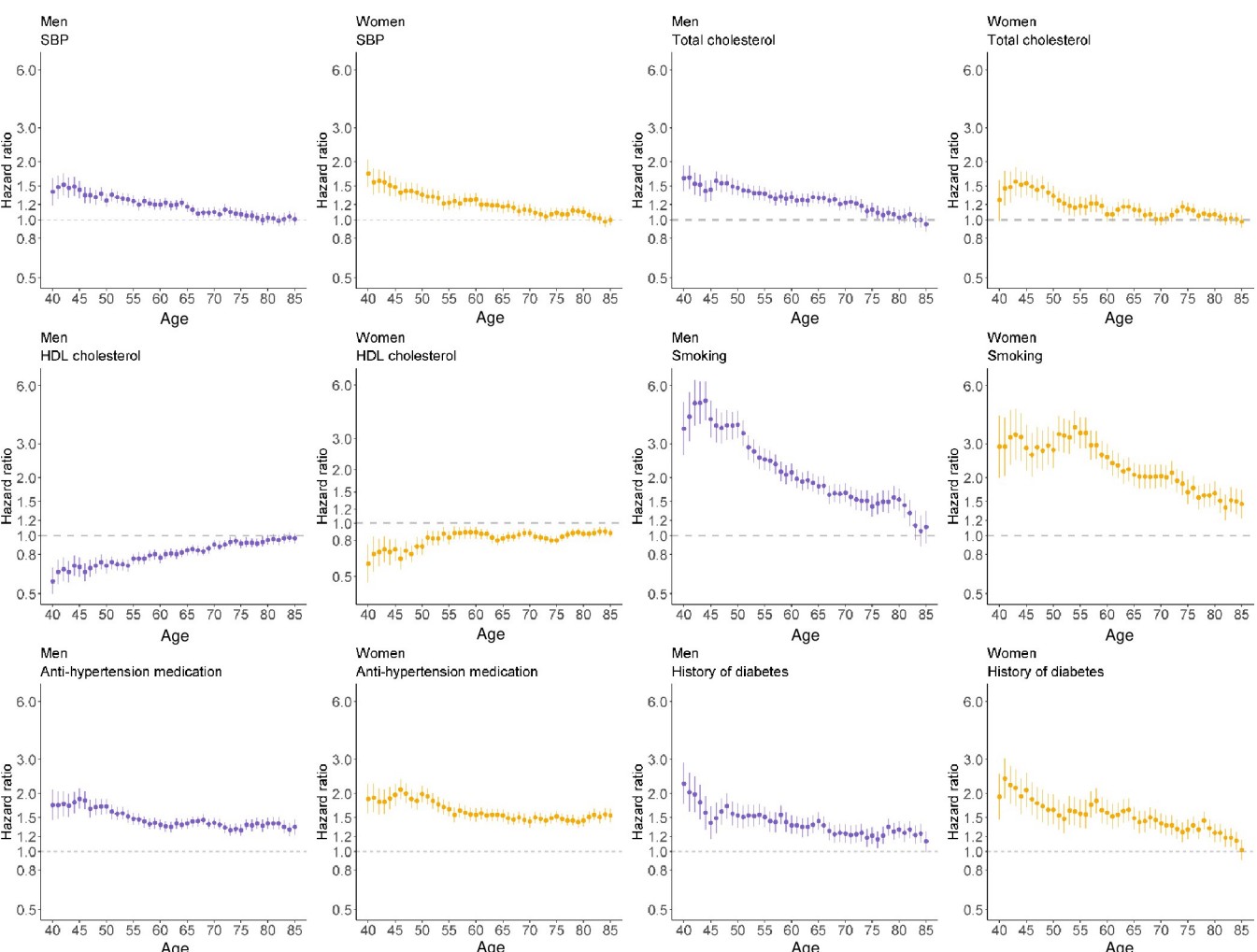

**Fig 2. Age and sex-specific hazard ratios of eHEART risk predictors with cardiovascular disease in derivation cohort.** Hazard ratios and 95% confidence intervals (shown as vertical lines) for association of cardiovascular disease with: systolic blood pressure, total cholesterol, HDL cholesterol, smoking status, anti-hypertension medication and history of diabetes by age, in men and women in the derivation dataset, Clinical Practice Research Datalink, Hospital Episode Statistics, and the Office for National Statistics, England, United Kingdom, 2004–2019. Hazard ratios are given per standard-deviation increase for SBP, total cholesterol, and HDL cholesterol. Hazard ratios and 95% confidence intervals are shown on the natural log scale.

## Population health modelling

Overall, 119,137 individuals from UK Biobank, with primary care records available, were used to model a representative population of 50,000 men and 50,000 women aged 40–69 in the UK (Fig 12 and Table 5 in S1 File). We estimated that there would be 3,566 and 1,819 CVD events over the next 10 years in men and women respectively. If QRISK2 was used as a formal assessment on the whole population, then 2,704 (75.8%) men and 895 (49.2%) women with CVD events over the next 10 years would be classified at high risk, ie QRISK2 ≥10% (Table 2). These percentages of "events captured" varied considerably by age group; for example, only 19% of events amongst 40–49-year-old men but 98% of events amongst 60–69-year-old men would be captured. Assuming statin therapy would be initiated for persons with predicted QRISK2 ≥10%, the number needed to screen to prevent one CVD event in men and in women would be 93 and 279 respectively. Assuming 50% invitation uptake, the number

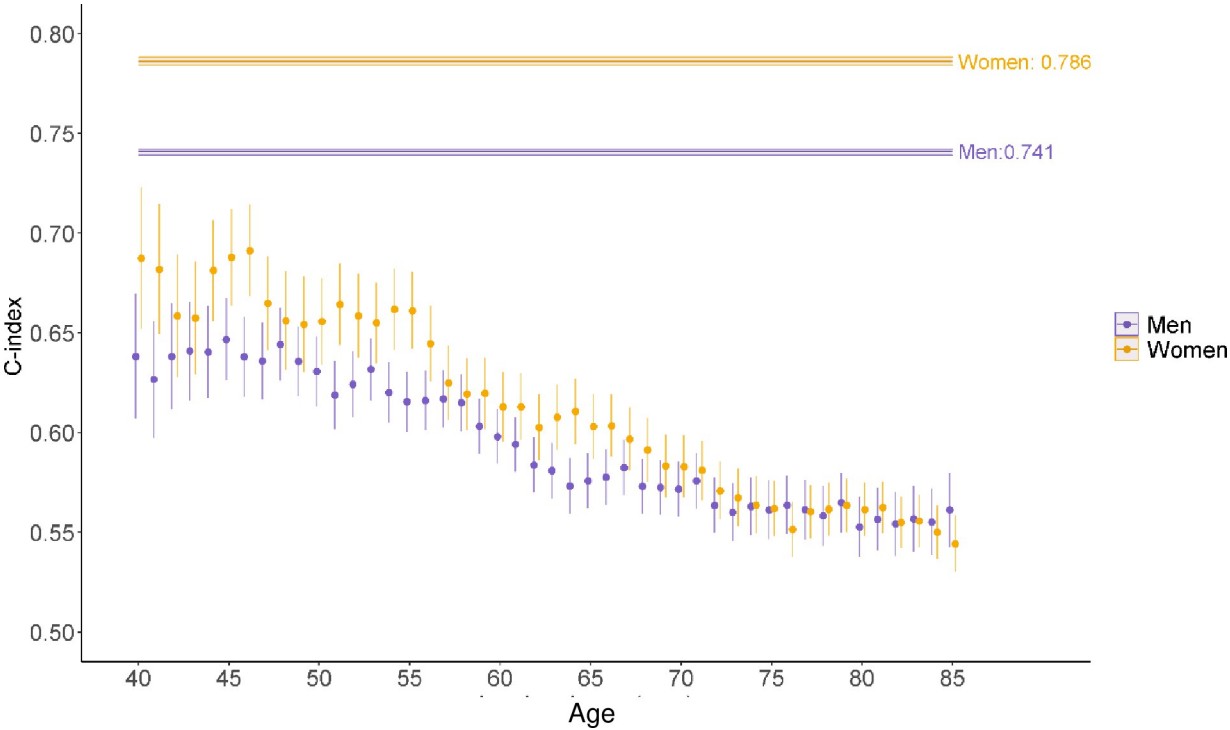

**Fig 3. Age and sex-specific C indices of eHEART in validation cohort.** C-indices and 95% confidence intervals (shown as vertical lines) from eHEART for the prediction of 10-year cardiovascular disease by age and for all ages, in men and women in the validation dataset, Clinical Practice Research Datalink, Hospital Episode Statistics, and the Office for National Statistics, England, United Kingdom, 2004–2019. The overall C-indices account for the discriminatory information in age, hence are higher than the age-specific C-indices.

needed to invite to prevent one CVD event in men and in women would be 185 and 559 respectively (Table 6 in S1 File).

If the eHEART tool with a prioritisation threshold of 10% was used to prioritise QRISK2 assessment in the population (and assuming all individuals had at least one primary care record of either smoking, SBP, HDL or total cholesterol) then 2,258 (63.3%) men and 573 (31.5%) women with CVD events over the next 10 years would be classified at high risk (Table 2). The number needed to screen to prevent one CVD event in men and in women would be 45 and 50 respectively. The number needed to invite to prevent one CVD event in men and in women would be 82 and 90 respectively (Table 6 in S1 File). In contrast, using age- and sex-specific prioritisation thresholds corresponding to 5% false negative rates would classify 2,616 (73.4%) men and 861 (47.3%) women with CVD events over the next 10 years as high risk. The number needed to screen to prevent one CVD event in men and in women would be 75 and 162 respectively. Assuming 55% invitation uptake after prioritisation, the number need to invite to prevent one CVD event in men and in women would be 136 and 294 respectively (Table 6 in S1 File).

## Sensitivity analyses

In comparison to eHEART being used as a prioritisation tool, if estimated QRISK2 with a prioritisation threshold of 10% was used to prioritise formal assessment in the population, then 2,409 (67.6%) men and 725 (39.8%) women with CVD events over the next 10 years would be classified at high risk. This equates to capturing approximately 87% of events when applying formal CVD risk assessment on the whole population (Table 7 in S1 File). The

**Table 2. Number needed to screen to prevent one event, and number of events captured when prioritising with eHEART in a hypothetical population of 100,000 individuals in England assuming all individuals invited for formal assessment attend.**

| Hypothetical population of 100,000 | | | | Full formal assessment only (QRISK2 threshold = 10%) | | Prioritisation tool (eHEART threshold = 10%) followed by full formal assessment (QRISK2 threshold = 10%) | | | Prioritisation tool (eHEART threshold corresponding to age- and sex-specific 5% false negative rates) followed by full formal assessment (QRISK2 threshold = 10%) | | | |
|---|---|---|---|---|---|---|---|---|---|---|---|---|
| Sex | Age group | Expected N | Expected CVD events in 10 years | Events captured N (%) | Number needed to screen to prevent 1 CVD event | Individuals prioritised N | Events captured N (%) | Number needed to screen to prevent 1 CVD event | Individuals prioritised N | Prioritisation threshold for eHEART | Events captured N (%) | Number needed to screen to prevent 1 CVD event |
| Men | 40–49 | 17673 | 516 | 96 (18.6%) | 1832.8 (1335.2, 2168.8) | 598 | 32 (6.3%) | 183.2 (76.5, 236.5) | 11054 | 3.0% | 91 (17.7%) | 1204.7 (849.1, 1435.1) |
| | 50–59 | 18061 | 1294 | 844 (65.2%) | 214.1 (204.3, 223.2) | 6348 | 518 (40.0%) | 122.7 (112.6, 131.0) | 14576 | 6.2% | 810 (62.6%) | 180.0 (170.7, 188.1) |
| | 60–69 | 14266 | 1756 | 1714 (97.6%) | 83.2 (82.6, 83.8) | 13391 | 1659 (94.5%) | 80.8 (79.8, 81.6) | 13391 | 10% | 1659 (94.5%) | 80.8 (79.8, 81.6) |
| | **Total** | **50000** | **3566** | **2654 (74.4%)** | **188.4 (185.2, 191.5)** | **20338** | **2209 (61.9%)** | **92.1 (90.4, 93.9)** | **39021** | **NA** | **2560 (71.8%)** | **152.4 (149.6, 155.1)** |
| Women | 40–49 | 17488 | 277 | 17 (6.1%) | 10605.6 (643.6, 14255.1) | 78 | 7 (2.5%) | 123.7 (0.0, 183.1) | 5495 | 2.0% | 16 (5.7%) | 3610.2 (0.0, 4933.9) |
| | 50–59 | 17986 | 607 | 116 (19.1%) | 1552.2 (1254.2, 1787.3) | 427 | 28 (4.5%) | 156.4 (75.4, 198.2) | 10519 | 3.3% | 113 (18.7%) | 930.5 (741.5, 1075.0) |
| | 60–69 | 14526 | 936 | 699 (74.7%) | 207.9 (201.1, 214.4) | 5174 | 437 (46.7%) | 118.4 (110.5, 124.6) | 11850 | 5.9% | 677 (72.4%) | 175.0 (168.7, 180.5) |
| | **Total** | **50000** | **1819** | **831 (45.7%)** | **601.6 (578.8, 622.7)** | **5680** | **471 (25.9%)** | **120.6 (113.1, 127.2)** | **27864** | **NA** | **805 (44.3%)** | **346.0 (332.1, 358.4)** |

Age structure of hypothetical population extrapolated from Office for National Statistics, England, United Kingdom 2017. Expected events at 10 years based on extrapolation of incidence rates from CPRD, 2014–2019. Age group and sex specific prioritisation thresholds were defined as the minimum of 10% and the level such that the expected false negative rate is controlled to be 5%. All individuals have at least one CVD risk factor (systolic blood pressure, smoking, total and/or HDL cholesterol) recorded for eHEART.

Number needed to screen based on assuming that all individuals are formally assessed. Under each scenario, some individuals are not formally assessed and may go on to have events. Statin compliance assumed to be equal to 50%.

number needed to screen to prevent one CVD event in men and in women would be 48 and 67, a reduction of 48% and 76%, respectively, when applying formal CVD risk assessment to the whole population. Notably, the events captured reduced by at least half amongst men under 50 years and women under 60 years. In contrast, using age- and sex-specific prioritisation thresholds corresponding to 5% false negative rates would classify 2,656 (74.5%) men and 871 (47.9%) women with CVD events over the next 10 years as high risk, around 98% events captured when applying formal CVD risk assessment on the whole population. The number needed to screen to prevent one CVD event in men and in women would be 73 (a reduction of 21%) and 148 (a reduction of 47%) respectively, with greatest reductions observed amongst younger individuals. The number need to invite to prevent one CVD event in men and in women would be 133 and 268 respectively (Table 8 in S1 File).

In sensitivity analyses including all individuals (i.e., all those without a primary care record for any one of SBP, HDL, total cholesterol and smoking status) (Table 9 in S1 File), we found

comparable results for eHEART and estimated QRISK2, but as expected, we observed greater numbers needed to screen especially among younger individuals (Tables 10, 11 in S1 File).

Comparable results were observed in additional analyses assuming a 5% (rather than 10%) formal risk assessment threshold (Tables 12 and 13 in S1 File). Using a fixed 5% prioritisation threshold resulted in a greater number needed to screen to prevent one CVD, than using a fixed 10% threshold for both prioritisation and formal assessment, however 88.5% of future CVD events in men and 63.3% in women being captured if prioritising with eHEART, and 91.3% of events in men and 71.8% of events in women being captured if prioritising with estimated QRISK2. Using age- and sex-specific prioritisation thresholds to limit the false negative rate to 2.5% resulted in comparable results for eHEART and estimated QRISK2.

## Discussion

We have provided quantitative evidence for systematically prioritising individuals for invitation for full formal CVD risk assessment using existing longitudinal primary care records, by evaluating a novel prioritisation tool (eHEART) with the use of age- and sex-specific prioritisation thresholds. Compared to conducting formal assessments on all individuals, the prioritisation tool has the potential to reduce the number needed to screen to prevent one CVD event by approximately 50% for women and 20% for men whilst identifying 96–98% of high-risk individuals. We found no added value of using the repeat measures in the eHEART prioritisation tool in comparison to estimates using single measures in QRISK2 as a prioritisation tool. The use of a fixed 10% prioritisation threshold substantially reduces the number of high-risk individuals identified by 10–20%. Our study highlights the importance of using prioritisation thresholds well below formal risk assessment thresholds, and lower than the 10% prioritisation threshold currently recommended by the NICE guidelines in the UK [7]. We demonstrated the advantage of using a pragmatic strategy of selecting age- and sex-specific prioritisation thresholds corresponding to 5% false negative rates, to ensure a balance between efficiency and specificity across age and sex.

This is the first study, to our knowledge, to demonstrate the benefits of age- and sex-specific thresholds for prioritising individuals for full formal CVD risk assessment, although they have been shown to improve specificity when applied directly to the formal CVD risk assessment [30, 31]. Note that whilst CVD risk prediction models already account for age and sex, they are not optimised to predict risk at any single threshold or for all subgroups of the population. Hence, there can be major differences in the specificity and sensitivity of CVD risk models used in combination with a threshold in different subgroups. In our study we specified the age- and sex- specific thresholds to minimise the false negative rate to 5% in each subgroup. However, alternative thresholds that optimise for a different criterion, such as a 1% or 10% false negative rate, could be chosen to achieve varying specificity and efficiency gains.

We have illustrated the use of prioritisation tools within the current United Kingdom population-wide preventative programme, in which all individuals aged between 40 and 74 years are invited into their general practitioners for a National Health Service (NHS) health check to assess an individual's risk of CVD, diabetes, kidney disease and stroke and dementia every five years [32]. A systematic review reported coverage (percentage eligible for health checks who received one) of 45.6% [26] and invitation uptake (percentage invited for health checks who received one) of 48.2% between 2009–2016 and suggested improved coverage and invitation uptake in populations targeted for prioritisation through general practices, including those with low socioeconomic status or ethnic minorities. During the COVID-19 pandemic, invited health checks in England declined by 82% between the end of 2019 and 2020 [33], leaving a concerning backlog [34–36]. As such, a complementary and automated systematic primary care records

based prioritisation approach could be supportive to improve both the coverage and uptake of the programme over the next few years, helping to reduce health inequalities [37].

This is also the first study to provide an independent evaluation of the use of the existing QRISK2 as a prioritisation tool compared with our novel eHEART tool. The derivation, validation and implementation of eHEART accounts for and leverages the sparse and sporadically observed longitudinal primary care records, by estimating current predictor values based on all past primary care records to make 10-year CVD predictions. By implementing a landmark model approach in deriving eHEART, the tool removes the need for complete risk predictor measurements in all individuals. Furthermore, eHEART uses only a small number of conventional risk predictors, thus making it transportable across different primary care electronic systems and country settings. In contrast, whilst the published derivation and validation methods for QRISK2 have used multiple imputation to handle non-recorded risk factors, its' clinical implementation is based on replacing missing non-recorded values with age, sex and ethnicity-specific population average values. However, QRISK2 contains many more risk factors than eHEART and is already integrated into the primary care electronic systems in England and used to prioritise patient invitation to formal CVD risk assessment. We conducted population health modelling utilising both primary care records and baseline measurements in UK Biobank, and used recalibrated risk estimates and observed incidence rates for estimating the population-health impact in England. Our results show only small differences between the tools, and that future focus should be on the selection and use of appropriate prioritisation thresholds.

Our study has potential limitations. First, we considered the QRISK2 rather than QRISK3 model, which includes risk factors such as disease history in 1st degree relative and stage of chronic kidney disease that are difficult to capture in electronic health records. However, we expect similar population health modelling results for QRISK2 and QRISK3, especially because of recalibration approach. Second, thresholds were selected by age-group and sex; alternative thresholds such as ethnic-specific thresholds could also be considered. Third, optimism may exist due to possible overlap of a small proportion of individuals in UK Biobank and in the databases used in deriving eHEART and QRISK2. Fourth, our population health modelling may be affected by healthy volunteer bias in UK Biobank; however, this is largely overcome by the recalibration and rescaling with more representative risk factor levels and incidence rates. Fifth, our population health modelling was restricted to individuals aged between 40–69 years due to data availability; we expect even larger efficiency gains if such a prioritisation tool was used in individuals <40 years but not >69 years. Sixth, we assumed a fixed level of statin compliance and uptake for all individuals; whilst these are likely to vary by age and sex [38–40], the levels of compliance and uptake are likely to change after prioritisation with age- and sex-specific thresholds is introduced. Seventh, our population health modelling focused on prioritising individuals for a formal CVD assessment at a single point in time, rather than every 5 or 10 years; however such prioritisation tools can be used in real-time with updated primary care records. Finally, our population health modelling was restricted to the population of England assuming use of the QRISK2 formal CVD risk assessment tool; future work should investigate the generalisability of the tools across different countries and in combination with other formal risk assessment tools, including QRISK3.

## Conclusions

The use of a prioritisation tool, based on either utilising repeated measures or last observed values of primary care records, in conjunction with age- and sex-specific thresholds has the potential to systematically identify high-risk individuals for invitation for full formal CVD risk assessment. Our results suggest using age- and sex-specific thresholds can reduce the number

needed to screen to prevent one CVD event whilst identifying the majority of high-risk individuals. Such a strategy could be particularly important as nations tackle the many COVID-19 related burdens and backlogs in primary care settings.

## Supporting information

**S1 File. Supplementary tables and figures.**
(PDF)

**S2 File. Codelists and supplementary methods.**
(PDF)

## Acknowledgments

CPRD uses data provided by patients and collected by the NHS as part of their care and support. This work was supported by Health Data Research UK, which is funded by the UK Medical Research Council, Engineering and Physical Sciences Research Council, Economic and Social Research Council, Department of Health and Social Care (England), Chief Scientist Office of the Scottish Government Health and Social Care Directorates, Health and Social Care Research and Development Division (Welsh Government), Public Health Agency (Northern Ireland), British Heart Foundation and Wellcome. For the purpose of open access, the author has applied a Creative Commons Attribution (CC BY) licence to any Author Accepted Manuscript version arising from this submission. This work was performed using resources provided by the Cambridge Service for Data Driven Discovery (CSD3) operated by the University of Cambridge Research Computing Service (www.csd3.cam.ac.uk).

## Author Contributions

**Conceptualization:** Emanuele DiAngelantonio, Ellie Paige, Juliet A. Usher-Smith, Angela M. Wood.

**Data curation:** Ryan Chung, Zhe Xu, Matthew Arnold, David Stevens, Angela M. Wood.

**Formal analysis:** Ryan Chung, Zhe Xu, Matthew Arnold, Ellie Paige, Angela M. Wood.

**Methodology:** Ryan Chung, Zhe Xu, Matthew Arnold, David Stevens, Ruth Keogh, Jessica Barrett, Lisa Pennells, Ellie Paige, Angela M. Wood.

**Software:** Ryan Chung, Matthew Arnold, David Stevens, Ruth Keogh, Angela M. Wood.

**Writing – original draft:** Ryan Chung, Zhe Xu, Matthew Arnold, Ellie Paige, Angela M. Wood.

**Writing – review & editing:** Ryan Chung, Zhe Xu, Matthew Arnold, David Stevens, Ruth Keogh, Jessica Barrett, Hannah Harrison, Lisa Pennells, Lois G. Kim, Emanuele DiAngelantonio, Ellie Paige, Juliet A. Usher-Smith, Angela M. Wood.

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
