## [Decision Letter · Decision Letter 0]

21 Jul 2023

PONE-D-23-05392Prioritising cardiovascular disease risk assessment to high risk individuals based on primary care recordsPLOS ONE

Dear Dr. Chung,

Thank you for submitting your manuscript to PLOS ONE. After careful consideration, we feel that it has merit but does not fully meet PLOS ONE’s publication criteria as it currently stands. Therefore, we invite you to submit a revised version of the manuscript that addresses the points raised during the review process.

We look forward to receiving your revised manuscript.

Kind regards,

Nikolas Pontikos

Academic Editor

PLOS ONE

Journal Requirements:

2. Please ensure that you have specified (1) whether consent was informed and (2) what type you obtained (for instance, written or verbal, and if verbal, how it was documented and witnessed). If your study included minors, state whether you obtained consent from parents or guardians. If the need for consent was waived by the ethics committee, please include this information.

“his work was supported by core funding from the: British Heart Foundation (RG/13/13/30194; RG/18/13/33946), BHF Cambridge Centre of Research Excellence (RE/13/6/30180) and NIHR Cambridge Biomedical Research Centre (BRC-1215-20014) [*]. *The views expressed are those of the author(s) and not necessarily those of the NIHR, NHSBT or the Department of Health and Social Care.”

“During the drafting of the manuscript, M.A. became an employee of AstraZeneca.”

We note that one or more of the authors are employed by a commercial company: AstraZeneca

4. PLOS requires an ORCID iD for the corresponding author in Editorial Manager on papers submitted after December 6th, 2016. Please ensure that you have an ORCID iD and that it is validated in Editorial Manager. To do this, go to ‘Update my Information’ (in the upper left-hand corner of the main menu), and click on the Fetch/Validate link next to the ORCID field. This will take you to the ORCID site and allow you to create a new iD or authenticate a pre-existing iD in Editorial Manager. Please see the following video for instructions on linking an ORCID iD to your Editorial Manager account: https://www.youtube.com/watch?v=_xcclfuvtxQ.

“This work was performed using resources provided by the Cambridge Service for Data Driven Discovery (CSD3) operated by the University of Cambridge Research Computing Service (www.csd3.cam.ac.uk), provided by Dell EMC and Intel using Tier-2 funding from the Engineering and Physical Sciences Research Council”

“This work was supported by core funding from the: British Heart Foundation (RG/13/13/30194; RG/18/13/33946), BHF Cambridge Centre of Research Excellence (RE/13/6/30180) and NIHR Cambridge Biomedical Research Centre (BRC-1215-20014) [*]. *The views expressed are those of the author(s) and not necessarily those of the NIHR, NHSBT or the Department of Health and Social Care.

This work was funded by the Medical Research council (MR/K014811/1). The study funders played no role in the design, analysis or interpretation of the study. R.C. is funded by a BHF PhD studentship (FS/18/56/34177). Z.X. is funded by the Chinese Scholarship Council. M.A. was funded by a British Heart Foundation Programme Grant (RG/18/13/33946). D.S. was funded by the NIHR Cambridge Biomedical Research Centre (BRC-1215-20014) [*]. J.B. was funded by a Medical Research Council fellowship (MR/L501566/1) and unit programme (MC_UU_00002/5). L.G.K. was funded by the NIHR BTRU in Donor Health and Genomics (NIHR BTRU-2014–10024) and is funded by the NIHR BTRU in Donor Health and Behaviour (NIHR203337) [*]. J.A.U. is funded by an NIHR Advanced Fellowship (NIHR300861). A.M.W. is part of the BigData@Heart Consortium, funded by the Innovative Medicines Initiative-2 Joint Undertaking under grant agreement No 116074. A.M.W. is supported by the BHF-Turing Cardiovascular Data Science Award (BCDSA\\100005).”

Additional Editor Comments (if provided):

Reviewers' comments:

Reviewer's Responses to Questions

**Comments to the Author**

1. Is the manuscript technically sound, and do the data support the conclusions?

Reviewer #1: Yes

2. Has the statistical analysis been performed appropriately and rigorously? 

Reviewer #1: Yes

3. Have the authors made all data underlying the findings in their manuscript fully available?

Reviewer #1: Yes

4. Is the manuscript presented in an intelligible fashion and written in standard English?

Reviewer #1: Yes

5. Review Comments to the Author

Reviewer #1: Thank you for allowing me to review the article. In this study, Chung et al. demonstrated using age- and sex-specific thresholds can lead to more efficient CVD assessment with eHEART. The study provides valuable insights into precision medicine as CVD differs among gender, and age is a significant contributor. The methodology was robust and well-written, and I enjoyed learning from it. However, I have some questions that I would like to discuss.

Why do you choose QRISK2, not the newer QRISK3 for comparison?

While all CVD risk prediction models such as Framingham, PCE, and QRISK3 account for age and sex, it is unclear why using age-sex thresholds still makes a difference. For instance, line 284-286 indicates that the percentages of “events captured” varied considerably by age group, with only 19% of events among 40-49-year-old men but 98% of events among 60-69-year-old men being captured. Even though QRISK2 also takes age and sex into account, why such significant differences are still observed?

Could it be possible that this improvement in prioritization resulted from recalibrating the algorithm on newer data that reflect the younger trend in CVD detection rate over recent years, whereas the data used for developing QRISK2 is old?

6. PLOS authors have the option to publish the peer review history of their article (what does this mean?). If published, this will include your full peer review and any attached files.

Reviewer #1: No

---

## [Author Response · Author response to Decision Letter 0]

5 Sep 2023

Amended funding statement 

This work was supported by core funding from the: British Heart Foundation (RG/13/13/30194; RG/18/13/33946), BHF Cambridge Centre of Research Excellence (RE/13/6/30180) and NIHR Cambridge Biomedical Research Centre (BRC-1215-20014) [*]. *The views expressed are those of the author(s) and not necessarily those of the NIHR, NHSBT or the Department of Health and Social Care.

This work was funded by the Medical Research council (MR/K014811/1). The study funders played no role in the design, analysis or interpretation of the study. R.C. is funded by a BHF PhD studentship (FS/18/56/34177). Z.X. is funded by the Chinese Scholarship Council. M.A. was funded by a British Heart Foundation Programme Grant (RG/18/13/33946). D.S. was funded by the NIHR Cambridge Biomedical Research Centre (BRC-1215-20014) [*]. J.B. was funded by a Medical Research Council fellowship (MR/L501566/1) and unit programme (MC_UU_00002/5). L.G.K. was funded by the NIHR BTRU in Donor Health and Genomics (NIHR BTRU-2014–10024) and is funded by the NIHR BTRU in Donor Health and Behaviour (NIHR203337) [*]. J.A.U. is funded by an NIHR Advanced Fellowship (NIHR300861). A.M.W. is part of the BigData@Heart Consortium, funded by the Innovative Medicines Initiative-2 Joint Undertaking under grant agreement No 116074. A.M.W. is supported by the BHF-Turing Cardiovascular Data Science Award (BCDSA\\100005).

Cambridge Service for Data Driven Discovery (CSD3) was provided by Dell EMC and Intel using Tier-2 funding from the Engineering and Physical Sciences Research Council (capital grant EP/P020259/1), and DiRAC funding from the Science and Technology Facilities Council (www.dirac.ac.uk).

There was no additional external funding received for this study

The funder provided support in the form of salaries for authors M.A., but did not have any additional role in the study design, data collection and analysis, decision to publish, or preparation of the manuscript. The specific roles of these authors are articulated in the ‘author contributions’ section.

Amended competing interests statement

During the drafting of the manuscript, M.A. became an employee of AstraZeneca. This does not alter our adherence to PLOS ONE policies on sharing data and materials.

Reviewer’s comments

Reviewer 1: Thank you for allowing me to review the article. In this study, Chung et al. demonstrated using age- and sex-specific thresholds can lead to more efficient CVD assessment with eHEART. The study provides valuable insights into precision medicine as CVD differs among gender, and age is a significant contributor. The methodology was robust and well-written, and I enjoyed learning from it. However, I have some questions that I would like to discuss.

 

Why do you choose QRISK2, not the newer QRISK3 for comparison?

Up until May 2023 the NICE guidelines recommended the use of QRISK2 in clinical practice. Since May 2023 QRISK3 is recommended. There are challenges in implementing the QRISK3 algorithm in research studies because it requires information on (i) stage of chronic kidney disease and (ii) angina or heart attack in a 1st degree relative, which are not well coded in electronic health records. For this reason, we retain our comparison against QRISK2. However, we expect similar population health modelling results for QRISK2 and QRISK3, especially because of our recalibration approach. 

We have justified our use of QRISK2 by adding the following text to the methods (lines 168-170): 

“We calculated formal 10-year CVD risks using QRISK2 (recommended by UK NICE guidelines until May 2023) using the values at the baseline assessment of UK Biobank’s and published coefficients.”

and added the following text to the limitations section in the discussion (lines 380-386 & 402-403): 

“First, we considered the QRISK2 rather than QRISK3 model, which includes risk factors such as disease history in 1st degree relative and stage of chronic kidney disease that are difficult to capture in electronic health records. However, we expect similar population health modelling results for QRISK2 and QRISk3, especially because of recalibration approach. “

“future work should investigate the generalisability of the tools across different countries and in combination with other formal risk assessment tools, including QRISK3. “

While all CVD risk prediction models such as Framingham, PCE, and QRISK3 account for age and sex, it is unclear why using age-sex thresholds still makes a difference. For instance, line 284-286 indicates that the percentages of “events captured” varied considerably by age group, with only 19% of events among 40-49-year-old men but 98% of events among 60-69-year-old men being captured. Even though QRISK2 also takes age and sex into account, why such significant differences are still observed?

All CVD risk prediction models are statistically optimised to predict 10-year CVD risk across the full distribution of risk and for all individuals in the training dataset. That does not mean they are optimised at any single risk threshold for all individuals, or for subgroups of individuals. It is also likely that some risk models are less optimal for certain subgroups because of lower numbers of events contributing to the derived model. 

Hence, we find differences occurring by age and sex groups at a single 10% threshold. 

We have added the following text in the introduction to clarify this point: 

“Note that whilst CVD risk prediction models already account for age and sex, they are not optimised to predict risk at any single threshold or for all subgroups of the population. Hence, there can be major differences in the specificity and sensitivity of CVD risk models used in combination with a threshold in different subgroups.”

Could it be possible that this improvement in prioritization resulted from recalibrating the algorithm on newer data that reflect the younger trend in CVD detection rate over recent years, whereas the data used for developing QRISK2 is old?

For the reason above, we believe that the key factor that contributes to the improvement in prioritisation is due to the use of the age-and sex specific thresholds. By comparing the impact of using a fixed 10% threshold, whilst keeping all other conditions fixed, we showed that the choice in risk thresholds has the largest impact on public health measures.

---

## [Editor Report · Decision Letter 1]

18 Sep 2023

Prioritising cardiovascular disease risk assessment to high risk individuals based on primary care records

PONE-D-23-05392R1

Dear Dr. Chung,

We’re pleased to inform you that your manuscript has been judged scientifically suitable for publication and will be formally accepted for publication once it meets all outstanding technical requirements.

Kind regards,

Nikolas Pontikos

Academic Editor

PLOS ONE

Additional Editor Comments (optional):

The authors seemed to have addressed the reviewer's comments.
---

## [Editor Report · Acceptance letter]

20 Sep 2023

PONE-D-23-05392R1 

Prioritising cardiovascular disease risk assessment to high risk individuals based on primary care records 

Dear Dr. Chung:

I'm pleased to inform you that your manuscript has been deemed suitable for publication in PLOS ONE. Congratulations! Your manuscript is now with our production department. 

Kind regards, 

on behalf of

Dr. Nikolas Pontikos 

Academic Editor

PLOS ONE